CBF transcription factors involved in the cold response of Camellia japonica (Naidong)

Fan Menglong
Zhou Rui
Liu Qinghua
Sun Yingkun sunyk678@qau.edu.cn
College of Landscape Architecture and Forestry, Qingdao Agricultural University , Qingdao , Shandong , China
Nogueira Fabio
Electronic publication date: 2021 Sep 9
Publication date: 2021
Volume: 9
Electronic Location ID: e12155
Received 2021 Apr 14; Accepted 2021 Aug 24
Copyright: ©2021 Fan et al.
Copyright year: 2021
Copyright holder: Fan et al.
License: This is an open access article distributed under the terms of the Creative Commons Attribution License, which permits unrestricted use, distribution, reproduction and adaptation in any medium and for any purpose provided that it is properly attributed. For attribution, the original author(s), title, publication source (PeerJ) and either DOI or URL of the article must be cited.
License URL: https://creativecommons.org/licenses/by/4.0/

Keywords: CBFs, C. japonica (Naidong), PacBio, Low temperature, Winter camellia, Abiotic stress

Funding: The Science & Technology for People’s Livelihood Program of Qingdao City No. 17-3-3-47-nsh The High Talents Scientific Research Fund of Qingdao Agricultural University No. 663-1113343 This study was funded by the Science & Technology for People’s Livelihood Program of Qingdao City (No. 17-3-3-47-nsh) and, the High Talents Scientific Research Fund of Qingdao Agricultural University (No. 663-1113343). The funders had no role in study design, data collection and analysis, decision to publish, or preparation of the manuscript.

==============================
CBFs belong to the ERF subfamily of the AP2 supergene family and often play an important role in the cold acclimation of temperate plants. However, the role of CBFs in Camellia japonica (Naidong), the only Camellia japonica population found in the temperate zones of China, remains unclear. It is very important to study the genetic composition of C. japonica (Naidong) to adapt to low temperature for Camellia species. Using full-length transcriptome data, we identified four CjCBF genes that respond to cold stress and analyzed their evolutionary relationships, domains, and expression patterns. The phylogeny of CBFs of 19 angiosperms divided the genes into three categories, and the four CjCBFs belong to a small subcluster. The strong response of CjCBF1 to cold treatment and its sustained high level of expression indicated that it plays an important role in the process of cold acclimation. A yeast two-hybrid assay revealed an interaction between CjCBF1, CjCBF2, and CjCBF5, and subcellular localization confirmed this finding. The expression of CjCBFs was tissue-specific: CBF1 was mainly expressed in leaves, and CBF3 was mainly expressed in stem. The responses of the four CjCBFs to drought and high temperature and the effect of light were also characterized. Our study provides new insight into the role of CBFs in the cold response in C. japonica (Naidong).

Introduction

Low temperature stress is an important abiotic stress that greatly affects the distribution of plants and social and economic development worldwide. The study of cold resistance mechanisms is becoming increasingly important as threats to the ecological environment grow. Plants can reduce the damage caused by low temperature through complex mechanisms, such as physiological metabolic pathways and molecular regulation (Agarwal et al., 2006a). Temperate plants can improve their cold tolerance by experiencing low, non-freezing temperatures (i.e., cold acclimation). This process involves stress perception, signal transduction, and the regulation of protein-coding genes, in which transcription factors act as important regulatory hubs that control a large number of genes to mediate the response to different stresses.

C-repeat binding factors (CBFs) belong to the ERF family of the AP2 supergene family, which has a common DNA-binding domain and two conserved motifs. They are distinguished by two signature sequences (PKK/RPAGRxKFxETRHP and DSAWR) (Haake et al., 2002). In some plants, CBFs have been shown to be key regulators in the cold acclimation pathway (Stockinger, Gilmour & Thomashow, 1997; Medina et al., 1999; Tayeh et al., 2013; Francia et al., 2016; Würschum et al., 2017) and are specifically associated with the C-repeat/dehydration-responsive motif (CRT/DRE; G/ACCGAC) of the promoter region of the COR gene (Lata & Prasad, 2011), which can mobilize a large number of genes involved in cold acclimation independent of the ABA pathway (Kodama et al., 2000; Graham et al., 1997; Ashraf & Foolad, 2007; Kornyeyev et al., 2001). CBF transcription factors have been shown to improve cold resistance in maize, potato, tomato, Arabidopsis, tobacco, and other plants through overexpression verification (Ito et al., 2006; Gilmour et al., 2000; Liu et al., 2011; Dubouzet et al., 2003; Mei et al., 2009; Al-Abed et al., 2007; Tang et al., 2011; Hsieh et al., 2002a; Hsieh et al., 2002b). In addition, previous studies have shown that CBF (DREB) not only participates in low temperature stress but also responds to drought and saline-alkali stress.

However, the overexpression of CBF transcription factors can inhibit plant growth and development and even lead to sterility (Tillett et al., 2012). This problem can be solved by incorporating different promoters into overexpression vectors with CBF genes of different species. For example, the incorporation of Arabidopsis DREB1A and rd29A into an expression vector was not observed to significantly inhibit growth and sterility (Kasuga et al., 1999). AtCBF1, AtCBF2, and AtCBF3 in A. thaliana are not only involved in the regulation of cold acclimation but also basic cold tolerance; however, AtCBF2 appears to be more important than AtCBF1 and AtCBF3 in the process of cold acclimation (Zhao et al., 2016). In Arabidopsis thaliana, CBF2 negatively regulates CBF1 and CBF3, but CBF1 and CBF3 do not directly regulate the expression of other CBFs; instead, they coordinate the regulation of some downstream target genes (Novillo, Medina & Salinas, 2007). Thus, studying the combined expression of multiple genes is essential for improving plant cold resistance.

Camellia plants are mostly distributed in the subtropics and are not tolerant of low temperatures. Temperature greatly affects the economic value of Camellia plants. C. japonica (Naidong), which is the most northerly distributed Camellia species in China, has excellent cold resistance, but molecular research has been limited because of a lack of genomic sequences. Transcriptomic study of C. japonica (Naidong) has shown that α-jasmonic acid may play an upstream signaling role in the response to cold stress (Li et al., 2016). However, no studies have identified CBF genes in C. japonica (Naidong) nor have characterized their expression patterns in the response to cold stress.

Here, we identified and isolated four low temperature-responsive CjCBFs transcription factors from low temperature treatment full-length transcriptome data (PacBio). Structure analysis, phylogenetic analysis, Gene Ontology (GO) annotation, and the expression patterns of genes were clarified by bioinformatics technology, and the role of CjCBFs in the response to low temperature stress was characterized in detail. Overall, this study provides valuable information that could be used to improve the cold tolerance of Camellia plants.

Materials and Methods

Identification of new CjCBFs genes in C. japonica (Naidong) involved in cold tolerance

We used our C. japonica (Naidong) low temperature and drought full-length transcriptome data based on PacBio and Illumina sequencing (PRJNA689105) as the local database. The protein sequences of CBF family genes (i.e., AtCBF1, AtCBF2, AtCBF3, AtCBF4) in Arabidopsis thaliana were queried to search the local databases using the BLASTP method with an E-value threshold of <1e−20. Then, HMMER3.0 (Potter et al., 2018) was used to further screen sequences using default parameters (E<1e−5), the AP2 domain was based on Pfam database (Pfam accession, PF00847). The longest protein was selected as a representative for the gene. Sequences that do not contain the complete AP2 domain and signature sequences were eliminated.

Sequence analysis

ExPASy (Gasteiger et al., 2003) was used to analyze the relative molecular weight, isoelectric point, hydrophilicity, and other basic information of CjCBFs according to default parameters.

Motif and phylogenetic analysis

MEME (http://meme-suite.org/tools/meme) was used for the conservedmotif analysis of CjCBFs and AtCBFs proteins, with the maximum number of motifs set at ten. The parameters were as follows: minimum motif width, 6; maximum motif width, 50; Selected the motif classic mode and 0-order model of sequences. MAFFT was used for sequence alignment according to default parameters (FASTA format, auto). The R package ggmsa was used to visualize the sequence alignment. In this study, C. japonica (Naidong) and 18 other plant species were selected from dicotyledons and monocotyledons, including Brachypodium distachyon, Oryza sativa and three additional monocotyledons, as well as Camellia japonica, Arabidopsis thaliana, Cucumis sativus and 19 other dicotyledons. The CBF-like protein data of the 19 species (with the exception of C. japonica) were downloaded from the NCBI Database. The protein sequences were selected by HMMER3.0. If the AP2 domain was truncated, or the AP2 domain match E-value exceeded 1e−5, the protein sequences were excluded. Based on this, 424 CBF-like genes were identified from the 19 plant species in order to construct the phylogeny tree. The phylogeny was constructed using the maximum likelihood method with 1,000 bootstrapping replicates in MEGA (v. 6.0) (Tamura et al., 2013). The R package ggtree (v2.0.4) was used to visualize the phylogenetic tree.

GO and interaction analysis

KOBAS (Xie et al., 2010) (http://kobas.cbi.pku.edu.cn/) and A. thaliana (thale cress) were used for the GO enrichment analysis. The sequences of four CjCBFs were inserted separately into pGADT7 Vector and pGBKT7 Vector; pGADT7-T+pGBKT7-Lam was the negative control and pGBKT7-53+pGADT7-T was the positive control. In order to verify self-activation, the pGADT7+ pGBKT7 - CjCBF1/ CjCBF2/ CjCBF3/ CjCBF5 plasmid was transformed into Y2HGold yeast competent cells, with 30 mg/ml α-gal employed to detect the self-activation. Furthermore, in order to determine the suitable concentration of AbA, the inhibitory self-activated AbA concentrations of 0.20 µl/ml, 0.30 µl/ml, 0.50 µl/ml, 0.70 µl/ml and 0.90 µl/ml were adopted for the five working solutions. The pGBKT7 and pGADT7 + CjCBF1, pGBKT7 and pGADT7 + CjCBF2, pGBKT7 and pGADT7 + CjCBF3, and pGBKT7 and pGADT7 + CjCBF5 plasmids were transformed into Y2HGold yeast competent cells, respectively, to exclude false positives. Moreover, the pGADT7+ CjCBFs and pGBKT7 + CjCBFs plasmid was transformed into Y2HGold yeast competent cells to verify the interaction between CjCBFs.

RNA-seq data

In order to explore the transcriptional expression of CjCBFs at low temperature stress across time, we evaluated our low temperature stress transcriptome (PRJNA689105) under a 4 °C treatment lasting 0 h, 12 h, 24 h, and 72 h, with four groups of transcriptome data. The expression of the CjCBFs gene was analyzed and the results were visualized by the R package pheatmap. qRT-PCR technology was employed to verify the transcriptome data and the co-expression method was used to predict the potential lncRNA regulating CjCBFs (Li et al., 2015).

Subcellular localization

Four DNA sequences encoding CBF proteins were inserted into the super1300-GFP overexpression vector, and Agrobacterium tumefaciens GV3101 containing super1300-CjCBF1, super1300-CjCBF2, super1300-CjCBF3, and super1300-CjCBF5 was injected into the Abaxial of tobacco (Nicotiana benthamiana) leaves (Li et al., 2013). A Leica DM2500-DM2500 LED fluorescence microscope was used to obtain fluorescence information.

Promoter cloning and Cis-Element analysis

C. japonica (Naidong) DNA was extracted via the CTAB method (Doyle & Doyle, 1987) to clone the promoter. The CjCBF1 upstream 5′ UTR sequence was obtained from the Takara Genome Walking Kit (Takara, Japan) following the manufacturer’s instructions. Design of nested PCR specific primers with primer 5.0 (Table S7). The obtained sequence was connected to a T vector for sanger sequencing and the Cis-Elements in the sequence were analyzed based on the PlantCARE database (Lescot et al., 2002).

Plant materials and low temperature treatment

C. japonica (Naidong) seeds were obtained from the Qingdao Botanical Garden (36°05′N, 120°08′E), Qingdao Agricultural University Cooperative, China, in September 2018. The seeds were sown in sand in the Qingdao Agricultural University culture room (20 ± 2 °C) during the winter until the fourth true leaf appeared. Following germination, the plants were transferred into the artificial climate room of the university (36°31′N, 120°39′E) under normal culturing conditions (plastic pots: top/bottom diameter 20/12 cm, height 26 cm; temperature: 25 ± 2 °C; relative humidity: 60%; soil: peat soil 80% + river sand 20%; natural light) for 3 months.

For the tissue specific expression experiment, we selected seedlings of a similar growth status and divided them into two groups, with three plants per group. Each group was treated at 4 °C for either 0 h or 24 h in a low temperature incubator. Tender roots, stems, leaves, buds, and flowers were collected at 0 h and 24 h. For the heat, cold and light treatments, seedlings were separated into several groups. The first group was placed at 40 °C (from 25 °C to 40 °C with a 1 °C increment per hour) and maintained for 24 h (2000lx light intensity and 60% humidity). The second group was placed at 40 °C (from 25 °C to 40 °C with a 1 °C increment per hour) and maintained for 24 h (60% humidity). The plants were wrapped with foil to avoid light. The third group was placed at 4 °C (from 25 °C to 40 °C with a 1 °C increment per hour) and maintained for 24 h (2000lx light intensity). The fourth group was placed at 4 °C (from 25 °C to 40 ° C with a 1 °C increment per hour) and maintained for 24 h. (1 degree per hour from 25 °C to 4 °C). The plants were wrapped with foil to avoid light. The drought treatment involved stopping watering for 20 days and measuring the relative soil water content to determine the drought degree (Table S8). Unstressed seedlings were used as the control samples (CK). Three biological replicates were collected from the same position at each collection time for RNA isolation.

RNA extraction and Expression analysis of qRT-PCR

The total RNA extraction was performed using the SPARKeasy A0305 kit (QingDao, China), following the manufacturer’s instructions. Each treatment comprised three replications. The integrity and concentration were detected by 1% agarose gel electrophoresis and a NanoDrop2000 (Thermo Fisher Scientific, Waltham, MA, USA). First-strand cDNA was prepared using the Takara PrimerScriptTMRT reagent kit with gDNA Eraser, and RT-qPCR primers were designed by Primer 5.0 (Table S7). Takara TB Green was used to set up the Premix Ex Taq qGreTli RNaseH Plus for the qRT-PCR experiment. Per the manufacturer’s instructions, 18S was used as an internal reference gene, and there were three biological replicates and three technical replicates in each group. The relative expression was determined by a three-step method (95 °C for 5 s, 55 °C for [15 s], and 72 °C for 30 s, repeated 40 times) on an Applied Biosystems StepOnePlus. The gene expression level was calculated according to the quantitative Ct method. The R package pwr was used to conduct Student’s t-tests. The R packages corrplot (0.84) and circlize (version 0.4.11) were used to conduct correlation analysis.

Results

Identification and characterization of CjCBFs

First, we used the transcriptome data of C. japonica (Naidong) as the local blast resource bank to obtain 11 candidate genes. Further screening incorrectly predicted CBF genes and redundant sequences, and four CBF transcription factors responsive to low temperature were identified. They were named according to their homology with CBF transcription factors in related plants (Camellia sinensis). The nucleotide and amino acid sequences are provided in Table S1. The full lengths of the CjCBF s sequences are 1.82 kb, 1.25 kb, 1.42 kb, and 1.35 kb. The CDSs are between 540 and 837 bp, and the number of amino acids are between 179 and 278. The molecular weights of the four cjCBF proteins are 30282.9 kDa, 19449.25 kDa, 26655.03 kDa, and 26526.23 kDa. The isoelectric point ranges from 4.9 to 9.8. Hydrophilicity ranges from 3.02 to 1.8 (Table 1).

Table 1 The physicochemical properties for the 4 CBF gene family members in C. japonica.

Gene name	Full length (kb)	CDS (bp)	Number of amino acids	Molecular Weight (kDa)	Theoretical pI	Grand average of hydropathicity	Aliphatic index	Instability index	
CjCBF1	1.82	837	278	30282.91	4.92	−0.533	66.08	53.31	
CjCBF2	1.25	540	179	19449.25	9.18	−0.411	74.19	51.14	
CjCBF3	1.42	741	246	26655.03	5.25	−0.407	66.71	46.86	
CjCBF4	1.35	720	239	26526.23	5.50	−0.328	74.35	48.40	

Analysis of conserved motifs

A comparison of the CjCBFs sequences (Fig. S1) revealed that the internal similarity of CjCBFs was 73.96%, and the domain sequence was highly conserved. Comparison of the domain sequences of A. thaliana and C. japonica (Naidong) (Fig. 1A) showed that the domain of C. japonica (Naidong) consists of the AP2 family universal DNA-binding domain (AP2 domain) and two signature sequences (DS-W-L), which were slightly different from the conserved domain of A. thaliana. We then analyzed the conserved motifs of A. thaliana and C. japonica (Naidong) and identified 10 motifs (Fig. 1B) with amino acid lengths between 6 and 50. In particular, Motif10 (WYQGDD) was the shortest (6 residues), while Motif1 (KKVRETRHPIYRGVRQRNSGKWVCEVREPNKKSRIWLGTFPTAEM AARAH) and motif2 (DVAAJALRGRSACLNFADSAWRLPIPESLCPKDIQKAA AEAAEAFRPELC) were the longest (50 residues) and contained the AP2 domain. Only CjCBFs have motif8 (YLDVSGDVAKP), and only A. thaliana has motif7 (TTDHGLDMEETLVE). Only CjCBF1 and CjCBF5 have motif10. The functions of most of the motifs have not yet been elucidated. The similarity of motifs within species indicates that CBFs within species are highly conserved. The composition of different conserved motifs may be related to variation in the functional differentiation of genes.

Figure 1 Multiple sequence alignment of CBFs gene and Conservative motif analysis.

(A) The AP2 domain is represented in the yellow box. (B) The architecture of the 10 conserved protein motifs predicted via MEME analysis. Each motif is represented in a different color (Motif 1–10).

Analysis of the phylogenetic tree

To characterize the evolutionary relationships of CBF s among species, we constructed a genetic tree (Fig. 2) using 424 CBFs from 19 species, including dicotyledons and monocotyledons. These genes were generally divided into three categories. Monocotyledons and dicotyledons were clearly separated, and dicotyledons were divided into two subclasses: Clade I and Clade II. However, some monocotyledons (e.g., oil palm) were divided into A-II subcluster in Clade I, and rice was also partially distributed in this subcluster. This cluster is close to Clade III, with groups containing only Monocotyledons. This may represent the region of functional overlap of dicotyledon and monocotyledon CBFs. This also indicates that this cluster of CBF ancestral genes had differentiated before the separation of monocotyledons and dicotyledons and thus that the function and pedigree of CBFs became gradually enriched. A-I and B-II were the largest subclusters in Clade I and Clade II, respectively. CjCBFs were divided into A-I subclusters and clustered with A. thaliana. CBFs were highly conserved within species, and CBFs of the same species often clustered together.

Figure 2 Phylogenetic analysis of CBFs.

Maximum Likelihood phylogenetic trees of the protein sequences of 424 genes encoding CBFs from 19 angiosperms. camellia (Cj: Camellia japonica) are marked red in the evolutionary tree, and Arabidopsis (At: Arabidopsis thaliana) are marked blue in the evolutionary tree. The other species are: Eucalyptus (Eug: Eucalyptus grandis); soybean (Gm: Glycine max); Cucumber (Cs: Cucumis sativus); Chili Peppers (Ca: Capsicum annuum); Cotton (Gh: Gossypium hirsutum); potato (St: Solanum tuberosum); western balsam poplar (Pt: Populus trichocarpa); Kiwi fruit (Ac: Actinidia Chinensis); Morning glory (Pn: Pharbitis nil); Sunflower (Ha: Helianthus annuus); Rubber Tree (Hb: Hevea brasiliensis); sesame (Si: Sesamum indicum); Carrot (Dc: Daucus carotavar.sativa); Medicago (Mt: Medicago truncatula); Brachypodium distachyon (Bd: Brachypodium distachyon); rice (Os: Oryza sativa); oil palm (Eg: Elaeis guineensis).

GO analysis

GO enrichment analysis of CjCBFs was conducted to better understand the biological function of CjCBFs. CjCBFs were enriched in nucleus (GO:0005634), DNA-binding transcription factor activity (GO:0003700), regulation of transcription, DNA-templated (GO:0006355), transcription regulatory region sequence-specific DNA binding (GO:0000976), glucosinolate metabolic process (GO:0019760), cold acclimation (GO:0009631), and response to cold (GO:0009409). The results of the GO analysis confirmed that CjCBFs play a role in cold acclimation (4 °C) (Fig. 3) (Xie et al., 2010).

Figure 3 GO analysis.

The GO enrichment results of CjCBFs based on KOBAs. The enrichment degree increases with the -log10 (p value) value.

Expression model of CjCBFs based on RNA-seq

Based on the data analysis of the full-length transcriptome under treatment with low temperature stress, the standardized FPKM cluster analysis heat map (Fig. 4A) shows that CjCBFs responded strongly to low temperature. Although CjCBF1 played a role in all three gradient treatment groups, it was significantly up-regulated at the 12 h stage, and the strength of the effect was significantly higher in the 12 h stage than in the 24 h and 72 h stages. The response of CjCBF3 to low temperature was basically the same in the three treatment groups. CjCBF2 responded in all three treatment groups but was significantly up-regulated in 24 h and then decreased in 72 h. The expression of CjCBF3 and CjCBF5 was similar to that of CjCBF1, except that CjCBF5 was significantly up-regulated in 24 h but then decreased significantly and remained expressed at a low level. qRT-PCR was used to verify the relative expression of CjCBFs. The pattern was nearly identical, indicating that our transcriptome results were reliable (Fig. 5C). The correlation analysis also supports this finding (Fig. 4B). There is much evidence showing that lncRNA participates in biological processes by regulating gene expression, including responses to stress in plants. Our SMART and next-generation sequencing data were used to predict possible lncRNA transcripts. A total of 147 possible CjCBF-containing transcripts (Fig. 4D) were predicted by lncRNA.

Figure 4 Expression analysis based on the transcript group data.

(A) Expression pattern of CjCBFs in response to cold 0 h(CK), 12 h(T1), 24 h(T2), 72 h(T3) treatments, The RNA-seq data were normalized based on the mean expression value (log2fold change, log2FC) of each gene, green and blue boxes indicate high and low expression levels, respectively. (B) Correlation analysis of four CjCBFs with different low temperature treatment groups (cyan region connects CjCBF1 and four treatment groups, yellow region connects CjCBF5 and four treatment groups, brown region connects CjCBF3 and four treatment groups, purple region connects CjCBF2 and four treatment groups), the wider the region, the higher the correlation. (C) Verification of the CjCBFs expression in the transcriptome via qRT-PCR. (D) Differential expression heat map of the lncRNAs that regulate CjCBFs based on the full-length transcriptome data. The data were normalized based on the mean expression value (log2fold change, log2FC) of each lncRNAs, green and blue boxes indicate high and low expression levels, respectively. Yellow, green and blue groups represent the regulation of CjCBF2, CjCBF1, and CjCBF5, respectively.

Figure 5 Protein interaction identification and subcellular localization.

(A) Self-activation verification. White colony indicates the ability of yeast to grow on the deficient medium, blue color indicates that the self-activating activity of CjCBFs was not able to grow on the deficient medium supplemented with AbA. This points towards the inhibition of the self-activating activity. (B) Yeast two-hybrid. Yeast cells exhibit normal growth on the nutrient-deficient medium, indicating the interaction between CjCBF1, CjCBF2 and CjCBF5. (C) Subcellular localization of Super1300, CjCBF1, CjCBF2 and CjCBF5 in Nicotiana benthamiana. In bright, the black line is the outline of the cell. For eGFP, the green region is the expression site of the plasmid with GFP fluorescent protein and the CjCBFs gene.

Interactions of CjCBFs

To further study the post-transcriptional regulation mechanism of CjCBFs under cold stress, we characterized the interactions among CjCBF1, CjCBF2, CjCBF3, and CjCBF5 using the Y2Hgold yeast two-hybrid technique. All CjCBFs were completely inhibited when the concentration of Aureobasidin A (AbA) reached 90 ng/µl, while the corresponding AbA concentration for CjCBF2, CjCBF3, and CjCBF5 was 30 ng/µl (Fig. 5A), The results of self-activation verification showed that the self-activation ability of CjCBF1 was significantly higher than that of other CjCBFs, CjCBF2-PGAD, CjCBF3-PGAD, and CjCBF5-PGAD were transformed into yeast Y2HGold cells in CjCBF1-PGBD; PGAD-T and PGBD-lam were transformed into the negative control, and PGBD-T and PGBD-53 were transformed into the positive control. The results of interaction verification showed that the yeast cells co-transformed with CjCBF1 and CjCBF2, CjCBF1 and CjCBF5 could grow normally on the nutrient-deficient medium, indicating that CjCBF1 could interact with CjCBF2 and CjCBF5 (Fig. 5B). To verify the yeast two-hybrid results, we analyzed the subcellular localization of CjCBF1, CjCBF2, and CjCBF5. CjCBF1 and CjCBF5 were located in the nucleus and cell membrane, and CjCBF2 was located in the cell membrane (Fig. 5C). These results were consistent with the yeast two-hybrid results. Several CjCBFs were expressed to aid the ability of Camellia japonica to cope with low temperature stress.

Expression of CjCBFs in five Camellia japonica tissues

To more fully characterize the spatial expression characteristics of CjCBFs in C. japonica (Naidong), the leaves, stems, flowers, buds, and roots of C. japonica (Naidong) were treated with 4 °C for 24 h. The expression of CjCBFs was detected by qRT-PCR. the results showed, In the normal cultured CK group, the expression of 4 CjCBFs was lower, but it was worth noting that the expression of CjCBF1, CjCBF3 in flowers was higher than that of CjCBF2, CjCBF5. This suggests that CjCBF1, CjCBF3 may be involved in flower development. after 24 h of 4 °C treatment, 4 CjCBFs were Significant difference, but only CjCBF1 was highly expressed in all five tissues, especially in the petals, and CjCBF2, CjCBF3, and CjCBF5 were expressed at low levels. The expression of CjCBF2 in roots and flowers was low, and the expression of CjCBF3 and CjCBF5 in stems, leaves, and buds was significantly higher, and the expression of CjCBF5 in buds was higher (Fig. 6). These results suggest that CjCBF1 may play an important role in the response to cold stress in C. japonica (Naidong).

Figure 6 Organ-specific expression of four CjCBFs.

(A–E) We measured the relative expression amounts of four CjCBFs in each tissue type for A, B, C, D, E, and set the expression amounts of CjCBF1 in the CK group to one for each tissue. This was then used to calculate the relative expression amounts of other CjCBFs for all tissue types. (F–I) We measured the relative expression of each CjCBFs in five tissues got F, G, H, I, and set the expression of each CjCBFs in the root of the CK group to 1. This was then used to calculate the relative expression of each CjCBFs in other tissues.

Bioinformatics Analysis of CjCBF1 Cis-Elements

Promoter cis-elements function as transcription factor binding sites and are key for the transcriptional regulation in response to abiotic stress (Moore et al., 2010). As CjCBF1 plays an important role in CjCBFs, we cloned and analyzed the 1,400 bp upstream sequences of CjCBF1. Bioinformatics analysis revealed the sequence to contain 31 cis-acting elements (Fig. 7 and Table S3), including 12 photosensitive responses and related cis-elements, CGTCA and TGACG motifs (involved in methyl jasmonate responsiveness). The sequence was also rich in TC-rich repeats related to stress, and MYB and MYC related cis elements. The results suggest that CjCBF1 has a wide response to abiotic stress and that light may play an important role in the expression of CjCBFs.

Figure 7 Prediction of cis-elements in the CjCBF1 promoters.

Expression of CjCBFs under different stresses and the effect of light

Camellia japonica (Naidong) was treated with high temperature stress, drought stress, low temperature stress, and light stress. Only CjCBF1 and CjCBF5 responded to high temperature (40 °C) stress (Fig. 8A), but the expression of CjCBF1 was significantly lower under 40 °C treatment than under 4 °C for 24 h and decreased slightly under joint high temperature and light stress treatment. Under low temperature and shading stress (Fig. 8B), the expression of CjCBFs decreased significantly compared with the non-shading treatment, which indicated that light may not represent a signal that CjCBFs respond to at high temperature; however, the effect of light on CjCBFs was more important at low temperature. The responses of CjCBF1, CjCBF3, and CjCBF5 to drought stress were weak; only CjCBF2 showed a significant response to drought stress, indicating that the responses of CjCBFs in Camellia japonica to drought stress are not strong (Fig. 8C).

Figure 8 Response of CBFs to multiple stresses.

(A) Relative expression of CjCBFs under the heat and hot (no-light) treatments. The expression of CjCBFs in each CK group was set to 1. (B) Relative expression of CjCBFs under the cold and cold (no-light) treatments. The expression of CjCBFs in each CK group was set to 1. (C) Relative expression of CjCBFs in the drought treatment. The expression of CjCBFs in each CK group was set to 1.

Discussion

When plants are subjected to abiotic stress, complex transcriptional and metabolic mechanisms are mobilized. Previous studies have shown that CBF transcription factors are essential for regulating responses to various stresses, and they can mobilize 35% of COR genes involved in cold acclimation independent of the ABA pathway. The ICE-CBF-COR pathway is widely known to be one of the core pathways in the response to low temperature stress in many plants, including A. thaliana. ICE1 and its homologous gene ICE2 dynamically regulate the expression of CBFs to ensure that they are expressed at appropriate levels in different periods. However, the regulation of CBFs is complex; for example, the promoter has at least one CRT/DRE motif COR gene, PHYTOCHROME-INTERACTING FACTOR4 (PIF4) and PIF7 (Barrero-Gil & Salinas, 2018) and the brassinosteroid signaling transcription factors BRASSIN AZOLE RESISTANT1 (BZR1) (Eremina, Rozhon & Poppenberger, 2016). In addition, the number of CBFs is often related to the size of the genome (Wang et al., 2019). Five CBF transcription factors have been identified in Camellia sinensis genomes, but the response patterns and characteristics of CBFs in C. japonica (Naidong) remain unclear. As the northernmost Camellia in China, exploration of the cold resistance mechanism of C. japonica (Naidong) is particularly important for enhancing the cold tolerance of Camellia. However, its genome was not sequenced. Using data from the PacBio and the Illumina full-length transcriptome, four CjCBFs were identified. These CjCBFs participated in the response to cold stress at non-freezing temperatures, and their expression was highest in the early stage of the stress response. They also responded to high temperature stress and drought stress.

Plants have evolved several strategies to cope with environmental stress. The evolution of multiple copies of genes can permit plants to quickly respond to stress. Multiple copies of CBF genes have often been observed (Zhao et al., 2016). However, there are no reference genome sequences of C. japonica (Naidong), precluding the possibility of determining the actual origin of these gene copies. In our transcriptome data, we detected multiple CBF transcripts, which have almost the same CDS region and significantly different UTR regions compared with ontological annotations. However, the expression levels of these copies were not the same. We speculate that the UTR region may also play an important role in the CBF pathway of cold-resistant Camellia japonica. Comparison with the database revealed that lncRNAs are not predicted in the UTR region of CjCBF; however, this does not mean that the UTR region is not involved in the response to low temperature stress. In the phylogenetic tree, CjCBF1 and CjCBF3 are clustered together, and CjCBF2 and CjCBF5 are clustered together. This, combined with the results of our yeast experiment, indicates that CjCBF1, CjCBF2, and CjCBF5 show different patterns of expression and are regulated differently. CjCBF3 appears to be a functional redundancy of CjCBF1 in the response to low temperature stress and does not directly participate in the internal interaction. These results confirm the core role of CjCBF1 in the response to low temperature stress. Several studies have shown that CBFs are not only involved in the stress response but are also involved in the inhibition of growth and development and delayed flowering. Analysis of the expression in five Camellia tissues in the absence of cold treatment revealed that CjCBF1 was highly expressed in flowers, but there was no significant change after cold treatment. Although CBF plays an important role in stress, CBF overexpression inhibits plant growth (Li et al., 2016; Zhao et al., 2016; Gilmour, Fowler & Thomashow, 2004); there is thus a need to find ways to avoid the negative effects of overexpression of these genes.

We obtained the upstream 1,400-bp sequence of CjCBF1 by the walking method (Table S2) and identified 12 homeopathic elements related to light, 1 cis-acting element involved in defense and stress responsiveness, and 8 cis-acting regulatory elements involved in MeJA responsiveness (Table S3). There are several light-response homeopathic elements in CjCBFs; We analyzed the response of CjCBFs to light under stress in detail through a light control experiment under low temperature. We found that the expression of CjCBFs decreased significantly under shading, indicating that light promotes the low temperature response of Camellia japonica.

However, light is not a necessary condition. Thus, the study of broad-spectrum and sensitive stress genes is particularly important and is the reason for increased interest in studying the effects of multiple types of stress. This suggests that the expression of CjCBFs is indeed different between day and night in winter. Interestingly (Gilmour, Fowler & Thomashow, 2004), our light control experiment under high temperature stress revealed that light did not cause changes in the expression of CjCBFs, and light intervention did not affect the response of CjCBF1 and CjCBF5 to heat stress. We found that low temperature induces the strongest and most complex regulatory response of CjCBFs. In addition, there are MeJA signal elements in the promoter sequence of CjCBF1, indicating that it may be regulated by jasmonic acid and may be involved in the metabolism of glucosinolates. CBF overexpression in A. thaliana promotes the accumulation of glucose, sucrose, and proline, which aids the survival of plants at low temperatures.

CBFs are structurally expressed in A. thaliana. CBF1 and CBF3 are the main components underlying the response to cold stress in plants (Gilmour et al., 2000; Harmer et al., 2000). CBF2 is involved in the regulation of CBF3 and CBF1. We verified the relationship between CjCBFs. CjCBF1 plays a central regulatory role. CjCBF1 may be expressed first and then regulate CjCBF5 and CjCBF2. The expression patterns indicated that the three CjCBFs were maintained in a dynamic balance, and the rapid up-regulation of CjCBF1 caused by low temperature breaks this balance. CjCBF2 and CjCBF5 negatively regulate CjCBF1 to maintain its level of expression within a specific range.

Conclusion

Four CjCBFs involved in the cold response of C. japonica (Naidong) were studied. An analysis of the motifs and evolution of these proteins provided information that could be useful for studying other Camellia species. The interaction mode of CjCBF1, CjCBF2, and CjCBF5 provided new insight into the biological function of CBF transcription factors in Camellia. The exploration of various stresses and responses to light expanded the known functions of CjCBF transcription factors and will aid future studies of the mechanism of stress adaptation in Camellia and other plants.

Supplemental Information

Supplemental Information 1 Sequence similarity analysis of CjCBFs

The white indicates that the structure of this position is not conservative. The color background indicates a conservative structure of the position.

Click here for additional data file.

Supplemental Information 2 Coding sequence and protein sequence of CBF members from camellia japonica(Naidong) and Arabidopsis

Click here for additional data file.

Supplemental Information 3 Promoter sequence of CBF1 from camellia japonica(Naidong)

Click here for additional data file.

Supplemental Information 4 The analysis of cis-elements of promoter

Click here for additional data file.

Supplemental Information 5 Raw data for Fig. 5C

Click here for additional data file.

Supplemental Information 6 Raw data for Fig. 7

Click here for additional data file.

Supplemental Information 7 Raw data for Fig. 8

Click here for additional data file.

Supplemental Information 8 The primer of qRT-PCR and promoter clon

Click here for additional data file.

Supplemental Information 9 The relative water content of soil

Click here for additional data file.

Additional Information and Declarations

Competing Interests

Author Contributions

Data Availability

The authors declare there are no competing interests.

Menglong Fan conceived and designed the experiments, performed the experiments, analyzed the data, prepared figures and/or tables, authored or reviewed drafts of the paper, and approved the final draft.

Rui Zhou performed the experiments, prepared figures and/or tables, and approved the final draft.

Qinghua Liu and Yingkun Sun conceived and designed the experiments, analyzed the data, authored or reviewed drafts of the paper, and approved the final draft.

The following information was supplied regarding data availability:

The data is available at NCBI GEO: PRJNA689105. The raw measurements are available in the Supplemental Files.

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
