# Peer review of "CBF transcription factors involved in the cold response of Camellia japonica (Naidong)"

_PeerJ, doi:10.7717/peerj.12155_

## Round 0.1 · original submission · Major Revisions

Dear Dr. Sun,

Thank you for submitting your manuscript. We have now completed review of your manuscript. As you can see from the comments below, two referees found your work interesting to publish in PeerJ. However, the second reviewer found several flaws in your experimental design.

We are willing to accept a new revised version of your manuscript, given that you are able to tackle all the raised concerns, mostly from reviewer 2.

Reviewer 1 ·

Basic reporting

The manuscript presents the characterization of four CBF genes identified in Camellia japonica and their functional analyses on different stress responses. The authors provide relevant study to explore the mechanisms of this species to adapt to stress conditions however, the manuscript lacks a large number of information that sometimes, but not always, they are in different parts of the text. There are statements missing references and the legends of the figures are totally incomplete making difficult to understand what the authors want to show. The manuscript requires a deep english review.

Experimental design

The manuscript lacks information on the methodologies and data of the experiments. Sometimes we find some information on the figures but they need to be provided in material and methods. In the discussion session, the authors describe information on the promoter region of the genes without any result be provided previously.

Validity of the findings

The results show novelties to the Camellia japonica species to explore the adaptation of plants to cold temperature.

Additional comments

I have some comments and suggestions to the authors as follow:

Abstract
line 13: review sentence: "..distributed in the northernmost edge of Camellia species in China, ..."

Introduction
Lines 50-51: requires reference. Perhaps this sentence of line 50-51 should be a new paragraph including the continuation with line 52, continuing.

Line 54: Requires a reference

Material and methods
Line 80: How many plants used in the experimental group and control group

Line 82: which method used to RNA extraction

Line 86: Confirm if the low temperature full-length transcriptome and drought full-length transcriptome database were both used. No information about the drought transcriptome were provided.

Line 87: Which technology was used to sequence the transcriptome? Is perhaps Illumina? (line 255 in discussion) please include also here.

Line 93: Include the references of the tools used for the analysis

Lines 95-102: Please, introduce the data that the phylogentic tree was prepared using 424 CBFs from 19 species. Also confirm if it was 424 or 324 as in Figure 3.

Line 100: Provide information of how the phylogenetic tree was constructed. It was on the basis of the similarities of the protein sequence of CBFs or by their conservative domains?

Line 107: Complete which plasmid was used to transform Y2HGold Yeast.

Lines 113-115: I am not convinced that transient expression is enough to show subcellular localization, provide references.

Line116-128: The description of the experiments and treatments should be described completely: temperatures used, treatments of drought stress, and which intensity of light. It is very critical how drought stress was measured in the experiments.

Results:
Lines 134-135: The CBFs were named to which related plants? Are they CjCBF1, CjCBF2, CjCBF3 and CjCBF5? we only see the names in figure 1.

Line 135: Please provide which supplementary file.

Line 141: The analysis shown on Figure 1 is not clear to demonstrate 73.96% of similarity between the 4 CjCBFs identified. Also, the legend of Figure 1 is not clear. I am not sure if the colors red, green, blue and purple represent the bases A, G, C , T or the 4 CBFs. Fig.1D shows CBF2, CBF3, CBF5 but in the legend also has CBF1. Please, be clear on that you want to show in the figure 1.

Figure 2. Please confirm if the colors represent amino acids or base? What do you mean of "same sequence of 5 or more"? "white" is the opposite to what? The legend of this figure is very confused.

Line 147: motif lengths between 6 and 50 amino acids? Please complete.

Line 156: The results related to the Phylogenetic tree are very confused. They genes were separated in clusters but the authors also mention about subclusters not shown in the figure. Please review.

Figure 3: The name of the 19 species used is missing in the legend. Confirm if it was used 424 or 324 amino acid sequences of CBF in the phylogenetic tree. I am not convinced that the R package ggtree was used to beautify the tree. Please clarify your statement.

Lines 159-160: How monocotyledons and dicotyledons were clearly separated from dicotyledons? I believe you want to say that monocotyledons and dicotyledons were clearly separated. And dycotyledons were divided into two subclasses. I see only Clade I, II and III in the figure 3, it is not shown the subcluster A-II in the Clade I.

Line 163: Could you please confirm if Clade III represents the large class of monocotyledons?

Line 166: Cluster A-I and B-II are in the clade I, II or III?

Line 170-177. I am not sure what you want to show in this analysis, the results in Figure 4 are not clear, what the different colors means?

The analysis of expression based on RNA seq is described based on figure 5. No in material and methods neither in this session, the treatments used were not described.

The reference Xie et al., 2010 describes the KOBAS 2.0: a web server for annotation and identification of enriched pathways and diseases. This tool was used to identify lncRNA? Please explain your goal in this analysis.

Line 200: AbA ou ABA?
Please provide description of these experiments.

Lines 214-225: The expression analyses in different tissues are shown in figure 7. Please describe group CK? as in the legend of Figure 7, the expression of CjDBF1 of root in CK group was set as 1, only for root? The relative expression of CjCBFs in Figure 7-E, F, G, H, I were related to CfCBF1? I believe the legend of this figure is completely nonsense and incomplete.

Lines 226-237: The expression of CjCBFs
Also the experiments were not described in the manuscript. I can not say if the treatment of high temperature was used


Supplementary materials: File S1, S2 and S3 listed on lines 318-321 do not correspond to the material sent. As supplementary materials there are 6 files:
the file S1: DjCBFs sequencing (xlsx);
file S2 shows the nucleotide sequence of the promoter sequence
File S3 shows the same file S2.
Files S4, S5, S6 are raw data for the qRT-PCR (xlsx).
Only in the discussion there is the analyes of those sequence related to the promoter region in the regulation of CBF expression. Base on which studies the authors claim they are promoter region?

Discussion
Line 248: Separate the words "brassinosteroid signaling"

Lines 249-250: "and the number of CBFs is often related to the size of the genome" could you please give references and examples about this statement?

Line 254: "the complex and large genome of Camellia" requires a reference

Line 283: The Supplementary table S2 shows the nucleotide sequence of the authors defined as promoter sequence of CBF1-C.japonica. These results may be important to be shown in the manuscript.

Line 285: The supplementary table S3 is missing

Lines 297-300: None of these studies were shown in the manuscript

Reviewer 2 ·

Basic reporting

Language needs to be improved.

Experimental design

Some data are confused and the experiments are not well designed.

Validity of the findings

no comment

Additional comments

In this study, the authors analyzed CBF genes in Camellia japonica. They identified four CjCBF genes, and all these genes are responsive to low temperature. The authors also analyzed the expression of CjCBF genes in different tissues after cold stress. Although the authors demonstrated the roles of CjCBF genes in cold stress response, some results in this manuscript are quite confused.


1. Line 132, what“11 transcripts”represents?
2. It is difficult to understand the pictures in Figure 1?
3. Figure 2B, how authors define different motifs? Are they known motifs?
4. Figure 5, what T1, T2, T3 mean?
5. It is not clear what Figure 5B means?
6. AbA is usually used for yeast-one hybrid, and 3-AT is used in yeast-two hybrid to inhibit autoactivation. Why authors used AbA in yeast-one hybrid assay?
7. Figure 6C, how can authors conclude that the CjCBFs are localized at plasma membrane?

Reviewer 3 ·

Basic reporting

Naidong is a geographical type of Camellia japonica distributed in the northernmost part in China, which cold tolerance is much higher than that of other C. japonica cultivars. Investigation on Its cold tolerance is of great significance to breeding of cold tolerant cultivars of C. japonica. The authors indentified four CBF genes in Naidong and analyzed these genes. The results provided new insight into the role of CBFs in the cold tolerance in C. japonica.

Experimental design

Naidong is one type of Camellia japonica that is distributed in the northernmost part of China. I think the expression of this material (Camellia japonica (Naidong)) in manuscript is not appropriate. Is it a geographical variation or a cultivar? There is no proper expression. Suggest to consider modification.
The description of experimental materials is too simple, and some information presented in the results is not expressed in this part, such as "40°C treatment" in LINE 229, "light stress treatment" in LINE 230, and "T1, T2, T3" in LINE 183.
The Latin names of some plants in manuscript are not italicized, so it is suggested to check and revise.

Validity of the findings

no comment

---

## Round 0.2 · Minor Revisions

Dear Dr. Sun,

Although your manuscript - CBF transcription factors involved in the cold response of Camellia japonica (Naidong) - is almost ready to be Accepted for publication, it still needs minor revisions before issuing the manuscript for publication. Please check the requested revisions below:

There are a number of small issues that need to be addressed before this is publishable.

1) Line 18: change “indicated” to “suggested”. This study does not prove the function of CjCBF1.

2) Figure 1: should be a Table

3) Line 91 and Figure 2: “Conservative” should be “conserved”.

4) Figure 2A: text too small, especially gene names.

5) Figure 5C: text too small

6) Figure 5A and 5D: what are the units? What does 0 represent? Why is a red/blue scale used for 5A and a red/green scale for 5D?

7) Figure 5B: this is impossible to understand. Try a different way of plotting these results, or explain better. One possibility would be “ggpairs” from GGally package

8) Figure 7: Text is of inconsistent size, e.g. compare 7E and 7F

9) Figure 9: Text is stretched horizontally. Also true to some extent in other figures, but severe here.

10) Figure 9 legend labels it as Figure 8

11) Line 132 “injected into the back of”. “Back” is imprecise. Abaxial? Adaxial?"

Looking forward to the new version of the manuscript.

Best regards

Fabio Nogueira

Reviewer 1 ·

Basic reporting

The manuscript was improved and my questions answered. I agree to the publication.

Experimental design

The complementation of the experiments was introduced into the manuscript.

Validity of the findings

The research is interesting for publication.

Reviewer 2 ·

Basic reporting

no comment

Experimental design

no comment

Validity of the findings

no comment

Additional comments

The authors have addressed all my concerns.

Reviewer 3 ·

Basic reporting

No comment

Experimental design

No comment

Validity of the findings

No comment

Additional comments

No comment

---

## Round 0.3 · accepted · Accept

Dear Dr. Sun,
Thank you for your submission to PeerJ.
I am writing to inform you that your manuscript - CBF transcription factors involved in the cold response of Camellia japonica (Naidong) - has been Accepted for publication. Congratulations!

This is an editorial acceptance; publication is dependent on authors meeting all journal policies and guidelines.

Next steps: Your article is being checked and you will receive a list of production tasks shortly. After you complete these tasks, your proofing PDF will be created (please do not proof check your reviewing PDF!).

Best regards

Fabio Nogueira